# A Rapid Review of Territorialized Food Systems and Their Impacts on Human Health, Food Security, and the Environment

**DOI:** 10.3390/nu13103345

**Published:** 2021-09-24

**Authors:** Gabrielle Rochefort, Annie Lapointe, Annie-Pier Mercier, Geneviève Parent, Véronique Provencher, Benoît Lamarche

**Affiliations:** 1Centre Nutrition, Santé et Société (NUTRISS), Institut sur la Nutrition et les Aliments Fonctionnels (INAF), Université Laval, Québec, QC G1V 0A6, Canada; gabrielle.rochefort.2@ulaval.ca (G.R.); annie.lapointe@fsaa.ulaval.ca (A.L.); annie-pier.mercier.2@ulaval.ca (A.-P.M.); Genevieve.Parent@fd.ulaval.ca (G.P.); veronique.provencher@fsaa.ulaval.ca (V.P.); 2École de Nutrition, Université Laval, Québec, QC G1V 0A6, Canada; 3Faculté de Droit, Université Laval, Québec, QC G1V 0A6, Canada

**Keywords:** territorialized food system, sustainability, human health, environment, food security

## Abstract

The global food system is facing multiple problems, including rising food insecurity, degrading environments, and an increased incidence of diet-related chronic diseases. International organizations are thus calling for a transition toward territorialized food systems to alleviate some of these challenges. Yet, limited evidence supporting the benefits of territorialized food systems is available. Our objective was to summarize the current body of literature on territorialized food systems and their impacts on human health, food security, and the environment using a rapid review methodology. Articles were retrieved from three databases and analyzed using keywords and inclusion criteria corresponding to territorialized food systems, environment, human health, and food security. Six relevant publications were identified. While this limited evidence suggests that territorialized food systems may have positive effects on all three dimensions, data are not consistent across publications. For example, territorialized food systems may contribute to improved diet quality, provide agroecosystem services, and contribute to food security. However, food produced within these food systems may have a higher carbon footprint and be less available than industrially produced food. This rapid review also highlights the siloed nature of the current research on territorialized food systems and emphasizes the need for more holistic and interdisciplinary research.

## 1. Introduction

The current global food system is facing multiple problems. There are two billion people suffering from food insecurity worldwide [1] while one-third of the food produced in the world is lost or wasted [2]. Food production capacities will need to increase considerably to feed the world’s growing population, which is projected to reach 8.5 billion by 2030 and 9.7 billion by 2050 [3]. The increased urbanization in many countries, with 60% of the world population predicted to live in cities by 2030, also threatens the population’s food security, with a larger urban food supply needed [4,5]. While the Millennium Development Goals decreased the number of undernourished individuals from 23.3% in 1990–1992 to 12.9% in 2014–2016 [6], there were still more than 690 million undernourished individuals in the world in 2019 [1] and the COVID-19 pandemic has aggravated the number of people facing food insecurity in developed and developing countries [7]. 

The problem of undernourishment often coexists with micronutrient deficiencies as well as overweight and obesity, and thus a large proportion of the world population suffers from some form of malnutrition. Low diet quality is responsible for micronutrient deficiencies in an estimated two billion people, while also contributing to the pandemic of overweight and obesity, which affects over 1.9 billion adults [8,9]. The global food system has favored the homogenization of diets and a rapid shift to unhealthy dietary patterns in many parts of the world, resulting in increased consumption of highly processed and poorly nutritious foods rich in sugar, salt, and fat, and of animal foods [10,11,12]. This transition of the traditional diet to Westernized, ready-to-eat diets has been paralleled by a rapid increase in the incidence of non-communicable diseases, such as cardiovascular diseases and diabetes, which are one of the largest causes of mortality worldwide [13,14,15].

In addition to contributing to food insecurity and societal chronic diseases, the current global food system poses major threats to ecosystems [16]. Agriculture accounts for up to 30% of global anthropogenic greenhouse gas emissions [17] and for 70% of global water withdrawals [18]. A globalized food system contributes to land degradation, deforestation, biodiversity loss, and eutrophication of waterways due to intensification of agriculture, simplification of agroecosystems, and the use of large amounts of fertilizers and pesticides [19,20,21]. Finally, the global food system faces multiples challenges caused by climate change, which threatens the sustainability of food production, and is considered to be the least resilient food system to face such disruptions compared to localized and diverse food systems [22,23].

In face of these unparalleled challenges, a transition toward more resilient and sustainable food systems is inevitable to guarantee equitable access to quality foods that are culturally acceptable and compatible with human and planet health [5,16,24,25,26]. The COVID-19 pandemic has exacerbated the shortcomings of a globalized food system [27,28,29] and international organizations, such as the Food and Agriculture Organization (FAO), have proposed that an orientation towards more food autonomy and local food systems constitutes a powerful lever to improve human and environmental health and enhance food security [30]. Territorialized food systems constitute a set of agri-food chains meeting the Sustainable Development Goals, i.e., a set of global goals to achieve a better and more sustainable future [31]. By definition, such food systems are located within a specific geographical area with a regional dimension and are coordinated by territorial governance [32,33]. However, to date, initiatives to build more sustainable local food systems suffer from a lack of scientific knowledge about the structures, interactions, dependencies, assets, constraints, and complex trade-offs that are specific to these systems [34]. Therefore, the objective of this rapid review was to synthetize the current body of literature on the potential impact of territorialized food systems on human health, food security, and the environment, and to identify gaps and areas for future research. More specifically, we aimed to evaluate and use the current available evidence to answer the following research question: How can a territorialized food system contribute to the food security and sustainable health of populations while protecting agricultural and food diversity as well as the environment?

## 2. Materials and Methods

### 2.1. Study Design

A rapid review methodology was used to synthetize evidence from studies and publications that focused on territorialized food systems, human health, food security, and the environment according to the recommendations established by the Cochrane Rapid Reviews Methods Group [35].

### 2.2. Eligibility Criteria

The research was limited to peer-reviewed studies and publications written in English and French. All types of study designs and publications were considered. Considering that the concept of a territorialized food system is relatively new, no restriction on the date of publication was included. To be eligible, studies and publications needed to assess any type of territorialized food system and address the impact of such a food system on three dimensions concurrently: human health, food security, and the environment. Studies and publications focused on national or global food systems were excluded. Theses, dissertations, commentaries, and perspective pieces were also excluded.

### 2.3. Literature Search

The literature search for this rapid review was conducted at a single point in time in February 2021 by the lead author (GR), as suggested by the Cochrane Rapid Reviews Methods Group [35]. The following electronic databases were searched to identify potentially relevant studies and publications for inclusion: PubMed, Web of Science, and CAB Abstract. Extensive searches of the gray literature and of publications not listed in the databases used were not conducted, consistent with the recommendations by the Cochrane Rapid Reviews Methods Group. The search strategy was developed with the help of an information specialist. A combination of keywords and medical subject heading (MeSH) terms related to food systems, territorialization, and food security were used. The concepts of human health and environment were not included in the search strategy per se but were used as inclusion/exclusion criteria during the selection of the publications. The search strategy and related search terms are provided in Table 1.

### 2.4. Study Selection

All retrieved titles and abstracts were screened by one member of the research team (GR) according to the inclusion and exclusion criteria. If necessary, a second member of the research team was consulted (AL). Full-text copies of papers judged to be potentially relevant to the review were retrieved. Using a standardized grid, the eligibility of all full-text publications was independently assessed by two members of the team (GR, AL).

### 2.5. Data Extraction

For each study and publication that met the inclusion criteria, data were extracted using a standardized grid by one member of the research team (GR). Correctness and completeness of the extracted data were checked by a second member of the team (AL). Information extracted included author, year of publication, type of study/publication, location, aim of the study/publication, data collection, type of territorialized food system studied, and relevant findings (i.e., human health, food security, and the environment). Study quality was not assessed because of the heterogeneity of the studies and publications retrieved.

## 3. Results

### 3.1. Search Yields

A total of 2894 articles retrieved from databases were screened for inclusion during the literature search. After duplicates were removed, 2207 articles were screened by title and abstract according to the inclusion criteria. During this stage, 2171 articles and publications were excluded if at least one of the following inclusion criteria was not met: being written in English or in French, assessing any types of territorialized food systems, or addressing the concept of human health and the environment. This led to full-text screening of 36 articles and publications, from which 30 were excluded because they did not meet any one of the following criteria: assessing a territorialized food system or assessing the impact of the food system on human health, food security, and the environment. As shown in Figure 1, six publications meeting our inclusion criteria (two literature reviews, one qualitative study, one mixed-methods study, one case study, and one descriptive study) were included in this rapid review [36,37,38,39,40,41]. Publications were from Canada (*n* = 1), Australia (*n* = 1), Kenya and Bolivia (*n* = 1), the Asia-Pacific region (*n* = 1), or not mentioned (*n* = 2). The characteristics of publications included in this rapid review are listed in Table 2.

### 3.2. Evidence Synthesis

#### 3.2.1. How Are Territorialized Food Systems Defined?

Different definitions of a territorialized food system were identified. For example, alternative commercial and civic food subsystems, as assessed by James et al. [38], were consider as territorialized food systems. These subsystems are associated with local or regional scales compared to industrial food systems, which operate at the global and/or national scale. More precisely, alternative commercial food systems comprise producer coops and community-supported agriculture and artisanal farms, while civic subsystems represent household and community gardens [38]. The use of short food supply chains (SFSCs) by farmers was also considered to represent a territorialized food system in the included articles. SFSCs meet the definition of a territorialized food system because they allow a relational and geographical proximity between the producer and consumer. They include farm stands, U-pick farms, box schemes and online sales to farmers’ markets, direct sales to supermarkets, and food hubs [36]. Two studies assessed territorialized food systems using the angle of neglected and undervalorised crops and species referred as development opportunity crops (DOCs) [37] and future smart foods (FSFs) [40]. Diversification of agriculture with the use of DOCs, such as minor grains and pulses, root and tuber crops, fruits and vegetables, and non-timber forest products, has the potential to support smallholder farmers and rural communities [37]. FSFs include cereals, roots and tubers, nuts and pulses, as well as other species that are nutrient dense, climate resilient, economically viable, and locally available or adaptable [40]. The literature search also identified home gardening and urban agriculture as another form of territorialized food system. Farming systems, such as home gardening, represents a territorialized food system because they procure fresh food to the household by combining different physical, social, and economic functions on land around the family home. Urban agriculture refers to agricultural production occurring within or around cities. It comprises hydroponic or aquaponic indoor production through sky farming, ground-based outdoor urban gardens and farms, rooftop gardens and farms, landscaping and nursery businesses, and urban livestock [39]. Finally, local, domestic indigenous, and agroecological food systems were also identified as territorialized food systems. These food systems are characterized respectively by short value chains, production and consumption of food within local households, and creation of network of producers, processors, and consumers for food production in and around a city [41].

#### 3.2.2. Can Territorialized Food Systems Contribute to Human Health?

Retrieved studies and publications suggest that territorialized food systems may contribute to improved nutritional status and health by diversifying and incorporating nutritious local crops (e.g., DOCs and FSFs) into the diet [37,40]. For example, integration of local crops, such as millet, lentils, chick peas, and others, in one’s diet may alleviate macro- and micronutrient deficiencies and non-communicable diseases associated with homogenized diets due to the high nutritional value of these foods [37,40]. Kahane et al. [37] suggested, however, that more education about the health and nutritional benefits of these foods is needed for people to integrate this practice into their daily dietary and consumption habits. Data also suggest that food producers, farming workers, and consumers perceive territorialized food systems as having a positive impact on their health as well as on achieving a healthier diet [41]. Lal has suggested in a narrative review that territorialized food systems may indeed contribute to a better diet quality and health by providing easy daily access to fresh fruits and vegetables [39]. One study included in this rapid review did not directly assess the link between the territorialized food system and health. However, the authors focused on healthy and sustainable diets and the availability of the foods that make up these diets within food systems (see below) [38].

It is important to stress that the present rapid review could not identify intervention studies that assessed the impact of territorialized food systems per se on diet quality or health. Furthermore, this rapid review reveals that the association between territorialized food systems and the wellbeing of farmers and the community has been poorly documented [36]. Specifically, farmers in the study by Mundler et al. reported that being part of a territorialized food system allowed them to develop “various skills”, although such skills were not detailed in the publication. They also found a higher proportion of women and of farmers with a higher level of education among those engaged in territorialized food systems compared to farmers engaged primarily in conventional food chains. Being engaged in a territorialized food system among farmers was also associated with better job satisfaction, more autonomy, ability to innovate, new learning opportunities, social recognition, and financial security. However, and contrary to expectations, territorialized food systems apparently did not result in better social cohesion between farmers and consumers.

#### 3.2.3. Can Territorialized Food Systems Contribute to Food Security?

Analysis and comments from three articles suggest that territorialized food systems can contribute to food and nutrition security by providing essential nutrients through diverse, sustainable, safe, and nutritious food and by their capacity to face unpredictable events [37,39,40]. Territorialized food systems have also been associated with food security in developed and developing countries and may play a significant role during crisis, war, and following disaster [37,39]. For example, it has been estimated that 15–20% of the world’s food supply is produced through urban agriculture. This provides a certain degree of autonomy to cities, which may become particularly important in a time of crises, such as the COVID-19 pandemic [39]. A study conducted in Kenya and Bolivia revealed that territorialized food systems received higher food security subscores than agro-industrial food systems in a framework assessing food sustainability based on five dimensions [41]. Mundler et al. [36] documented initiatives taken by farmers engaged in territorialized food systems to improve access to SFSCs and educational activities among disadvantaged populations as a means to alleviate food insecurity. They also reported that food prices in territorialized food systems were not higher than in conventional stores. In contrast, James et al. [38] showed that territorialized food systems were less available in low-income neighborhoods and that food items consistent with a healthy and sustainable diet were also less available within these food systems.

#### 3.2.4. Can Territorialized Food Systems Contribute to Protecting the Environment and Improve Climate Resilience?

Three studies reported that territorialized food systems may provide many agroecosystem services, such as biodiversity, soil fertility, agroforestry, microclimate moderation, water quality, and control of run off and inundations [39,40,41]. Jacobi et al. found that territorialized food systems in Kenya and Bolivia have a lower carbon footprint, are less reliant on the use of external inputs, and allow for better and more efficient recycling of organics matters compared to agro-industrial food systems [41]. The analysis by Mundler et al. also revealed that the use of sustainable agricultural practices, such as organic farming, presence of windbreaks, green manure, winter cover crops, lower use of pesticides and fertilizers, and increased agrobiodiversity, was more prevalent in farms integrated into a territorialized food system than in conventional farms [36]. However, James et al. [38] found that territorialized food systems do not always have a lower carbon footprint as the type and volume of food produced by such systems need to be factored in. For example, they found that chicken produced within a territorialized food system had a higher carbon footprint than industrially produced chicken, due to the fact that industrial chicken production uses less land and less feed per chicken, which also have a shorter life span. On the other hand, lettuce produced within a territorialized food system had a lower carbon footprint than industrially produced lettuce. The analysis by James et al. also revealed that environmental concerns were not a priority for consumers when purchasing food, emphasizing the importance of implementing strategies that will be successful in convincing the consumer to participate and adhere to more sustainable food systems.

Territorialized food systems through cultivation of diverse local crops (e.g., DOCs and FSF) do seem to be more climate resilient, sustainable, and adapted to environmental stresses, although more research is needed on this topic [37,40]. However, evidence also suggests that different food subsystems (i.e., food systems operating at a local, regional, and global scale) are required to increase climate resilience [38]. The resilience of territorialized food systems, such as agroecological and local food systems, but not domestic-indigenous food systems was found to be higher than the resilience of agro-industrial food systems. Food system resilience was defined as the self-organization, buffer capacity, and capacity for learning and adaptation of the food system and assessed by different ecological as well as social indicators [41].

## 4. Discussion

A territorialized food system, which is defined as a set of agri-food chains meeting the Sustainable Development Goals [32,33], theoretically addresses food security as well as the sustainable health of the population and of the environment. This rapid review identified very few publications that discussed the concurrent associations between territorialized food systems, human health, food security, and the environment, thereby exemplifying the siloed nature of the current scientific research on food systems, as reported elsewhere [38,42]. Indeed, research on food systems and particularly on alternative food systems in the last decade has traditionally focused on only one singular sustainability issue [43]. This has contributed to an important knowledge gap on the interrelations between human health, food security, and environmental health across the food systems [44]. A more interdisciplinary, holistic, and systemic approach to the study of food systems is urgently warranted to ensure the transition toward sustainable food systems that are optimized and adapted to address all dimensions of human and environmental health [22,24].

Four of the six publications included in the rapid review provided insights regarding the nutritional benefits of foods produced within territorialized food systems, with a particular focus on essential nutrient procurement and food freshness [37,39,40,41]. However, no study identified in this rapid review assessed how procurement of foods through territorialized food systems influenced the overall diet quality, eating habits, or health variables and outcomes. Observations from the available studies mostly remained theoretical with no data from intervention studies. A few studies have assessed the associations between consumption of local food and diet quality, but as reported elsewhere, very few of these studies have used valid and reliable dietary assessment tools to assess the impact of consuming food through local food chains on nutrition-related outcomes [45]. Moreover, there is limited evidence regarding the wellbeing of farmers and consumers engaged in territorialized food systems, with only one study identified in the present rapid review suggested potential benefits among famers but not among consumers [36]. The lack of benefits among consumers was counterintuitive considering that territorialized food systems theoretically allow for the creation of a relation based on trust and social cohesion [43,46] and has previously been associated with psychological wellbeing [47]. Thus, additional research is needed to better understand and document how territorialized food systems may impact on the wellbeing and health of its many stakeholders.

Most of the studies we reviewed suggested that territorialized food systems can contribute to food security in developed and developing countries [37,39,40,41]. However, findings regarding the four dimensions of the concept of food security defined by the FAO [48], i.e., physical availability to food, economic and physical accessibility to food, adequate use of food, and the stability of those three dimensions over time, were either unconvincing, inconsistent, or contradictory. In particular, there is a lack of robust evidence regarding the dimension related to economic and physical accessibility to food. Physical accessibility to food produced within territorialized food systems can in fact be a barrier to many individuals when, for example, a visit to a far-located farm stand is required to access such foods [36,49]. Additionally, although food price may not be systematically higher in territorialized food systems [36], it has been reported elsewhere that food in local chains tends to be less affordable than food in global chains [50] and are perceived as more expensive by non-users [46]. Results also suggested that healthy and sustainable food items were less available in territorialized food systems [38]. This somewhat contradicts the observations that local food systems expose consumers to more healthy food, such as vegetables and fruits, at an affordable price [51]. Our review suggested that territorialized food systems may have the capacity to face multiple sanitary, climatic, social, or economic crises [37,39]. However, the current role of territorialized food systems in relation to an external crisis has to be evaluated more deeply [52] since positive and negative effects have been observed during the recent crisis [29]. In sum, more research is required as very few studies to date on food security were designed to capture the availability, accessibility, utilization, and stability of food produced through territorialized food systems.

The studies we reviewed suggested that territorialized food systems may contribute to environmental protection and provide many agroecosystem services that directly or indirectly benefit humans or improve social welfare [39,40,41]. The increased biodiversity that generally comes with territorialized food systems was one of the most reported environmental benefits of the studies reviewed [36,39,40,41]. This aligns with findings suggesting that local food chains are associated with more biodiversity than global chains [22,50]. However, the way in which territorialized food systems provide these agroecosystem services was not clearly described and some of the studies reviewed did not use any measurement to support these statements. Although it is assumed that alternative and local food systems are inherently good for the environment due to their very nature [43], very limited evidence from empirical studies supports these purported benefits [53,54]. Findings regarding the carbon footprint of territorialized food systems were somewhat contradictory and inconsistent. This is in part due to important differences in the measurement of this indicator (i.e., assessing different types of food through one or more activities along the value chain), to variability in the geographical area where the food is grown, and to differences in the methods used [44]. The results of our rapid review also highlight the need for additional research to determine which food system is most climate resilient. Moreover, since environmental preoccupations may still have little influence on consumers when purchasing food [38], we need to better understand how to educate people about the environmental consequences of the food they buy and eat in order to support the demand and development of more sustainable local food systems [23].

## 5. Conclusions

As indicated above, the objective of this rapid review was to answer the following research question: How can a territorialized food system contribute to the food security and sustainable health of populations while protecting agricultural and food diversity as well as the environment? Although the territorialization of food systems is strongly recommended to address the numerous challenges related to human and environmental health, this rapid review highlighted the fact that there is currently very limited pragmatic evidence to support this paradigm and the evidence available to date remains inconclusive in many aspects. Furthermore, most of the available evidence is limited by the observational nature of the research, with very few to no intervention studies performed to date. This rapid review also reveals that very few studies have assessed the potential impact of territorialized food systems on the three following sustainability dimensions concurrently: human health, food security, and the environment. This calls for more research in this area, with particular efforts towards integrating all health and environmental dimensions of territorialized food systems. There is a compelling need to decompartmentalize the empirical research on territorialized food systems, which by tradition is siloed in nature. Interdisciplinary and participatory approaches have already been initiated by Canadians researchers to better understand the local food system, such as the Université Laval (Quebec City area) [55] and the Laurier Centre for Sustainable Food Systems (Waterloo aera) [56], but further intervention research integrating human health, food security, and the environment is still needed. Transforming food systems to ensure food security and the sustainable health of populations while protecting agricultural and food diversity as well as the environment will require the collaboration of experts and scientists from a broad range of disciplines.

## Figures and Tables

**Figure 1 nutrients-13-03345-f001:**
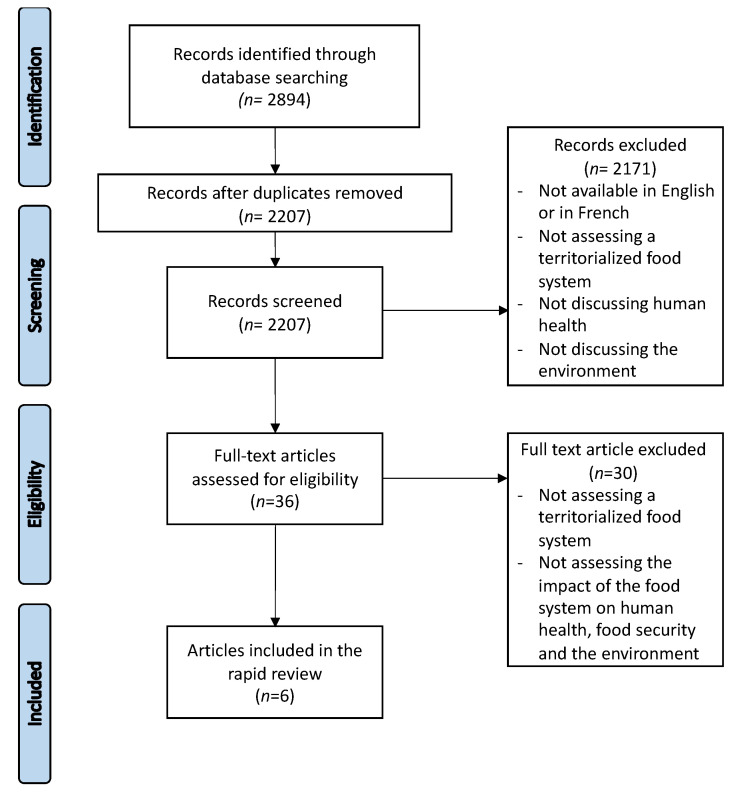
Flow diagram of the article selection process.

**Table 1 nutrients-13-03345-t001:** Key search terms used in academic databases to identify the literature relevant to food systems, territorialization, and food security.

Key Terms: Food System	Key Terms: Territorialization	Key Terms: Food Security
“food system” OR “food systems” OR “food hub” OR “agri-food system”	territorialization OR local OR rural OR regional OR urban	Diet quality OR food security OR food insecurity OR food insecurities OR food supplies OR food supply OR healthy diet OR healthy diets OR healthy eating index OR Healthy eating indices OR feeding behaviors OR eating behavior OR eating behaviors OR feeding patterns OR feeding pattern OR food habits OR food habit OR eating habit OR eating habits OR dietary habits OR dietary habit OR diet habits OR diet habit OR nutritional deficiency OR nutritional deficiencies OR undernutrition OR malnourishment OR malnourishments OR food quality OR food qualities OR nutritive values OR nutritive value OR nutritional biologic availability OR nutritional value OR nutritional values OR nutrition value OR nutrition values OR nutritional availability OR nutritional quality OR nutritive quality

**Table 2 nutrients-13-03345-t002:** Publications included in the rapid review of territorialized food systems (*n* = 6).

First Author and Publication Year	Study Design, Publication Type	Location	Aim	Data Collection Measures	Territorialized Food System Studied	Relevant Findings, Observations, Conclusions
James, 2015 [38]	Observational, mixed-methods study	Greater Western Sydney, Australia	To determine key points of intervention in urban food systems to improve the climate resilience, equity, and healthfulness of the whole system.	Life-cycle analysis to assess environmental footprint. Availability, affordability, and accessibility of healthy and sustainable diet assess in five socio-economic neighborhoods surveying 82 food provisioning outlets. Interviews with households in three different socio-economic areas.	Food systems operating at local and regional scale: Alternative commercial (e.g., producer coops, community supported agriculture, artisanal farms) and civic (e.g., household and community gardens) subsystems.	Food security: Availability of food items consistent with a healthy and sustainable diet is limited in alternative commercial and civic subsystems. Environment: Industrial subsystem can be more environmentally sustainable than civic subsystem. There is a need to consider the type of food and the volume produced when assessing the impact of a subsystem on the environment.
Mundler, 2016 [36]	Observational, qualitative study	Quebec, Canada	To build a systemic analysis model on the benefits attributed to short food supply chains (SFSCs) in order to measure their contribution to territorial development.	Questionnaire to farmers, qualitative interviews with stakeholders and farmers and price surveys.	Food supply chains allowing a relational and geographical proximity: short food supply chains (SFSCs) (e.g., farm stands, U-pick farms, box schemes and online sales to farmers’ markets, direct sales to supermarkets and food hubs).	Human health: SFSCs allow the development of various skills. There is a higher level of education among farmers, higher presence of women and job satisfaction. SFSCs do not create a social cohesion. Food security: Initiatives have been taken to improve access to SFSCs, and prices are not higher than those in conventional stores. Educational activities for consumer are promoted by farmers. Environment: More certified organic farming, presence of windbreak, green manure and winter cover crops in farms engaged in SFSCs. Lower use of pesticides and fertilizers and more varieties grown.
Kahane, 2018 [37]	Narrative literature review	-	To review science-based evidence arguing that diversification with greater use of highly valuable but undervalorised crops and species should be an essential element of any model for sustainable smallholder agriculture.	-	Neglected and undervalorised crops and species with a great potential to support smallholder farmers and rural communities: Development opportunity crops (DOCs) (e.g., minor grains and pulses, root and tuber crops, fruits and vegetables and non-timber forest products).	Human health: Locally available DOCs can be used to improve nutrition and health of rural and indigenous communities. Research is needed on the health and benefits of those crops. Food security: DOCs can contribute significantly to food security and nutrition at local and regional levels. Use of crop diversity is linked to food security in many countries. Environment: Complex agroecosystems with DOCs promote resilience and ecosystem functionality. More research is needed on crops diversity and local species.
Li, 2020 [40]	Observational, descriptive study	Asia-Pacific region	To demonstrate the multidimensional benefits of future smart foods (FSF) as an effective means to bridge production and nutrition gaps to address Zero Hunger.	Stage 1: scoping and identification of neglected and underutilized species. Stage 2: validation and prioritization of neglected and underutilized species. Stage 3: mapping using geographic information system.	Neglected and underutilized species that are nutrient dense, climate resilient, economically viable, and locally available or adaptable: Future smart foods (FSF).	Human health: FSF can improve diet quality and address micronutrient deficiencies and non-communicable diseases due to their nutritional qualities. Food security: FSF can contribute to food security by providing essential nutrients.Environment: FSF contribute to crop diversification, have many environmental benefits, and enhance the resilience and sustainability of the food system.
Lal, 2020 [39]	Narrative literature review	-	To describe home gardening and urban agriculture (HGUA) for food and nutrition security and ecosystems services provisioned by HGUA.	-	Farming system combining different physical, social, and economic functions on land around the family home: Home gardening. Agricultural production occurring within or around cities: Urban agriculture (e.g., urban gardens and farms, hydroponic or aquaponic indoor production, rooftop gardens and farms, landscaping and nursery businesses, and urban livestock).	Human health: HGUA provide easy access to fresh food and contribute to a better nutrition, human health, and wellbeing. Food security: HGUA can improve food and nutrition security by providing diverse, sustainable, safe, and nutritious food. Environment: HGUA strengthen many ecosystems services (e.g., biodiversity, microclimate moderation, water quality, and control of run off and inundations).
Jacobi, 2020 [41]	Observational, case study	Bolivia and Kenya	To analyze and compare the sustainability of six food systems in Kenya and Bolivia.	Surveys, structured and semi-structured interviews, focus group discussion, 24-h memory of what family consumed, direct and participant observation, life-cycle inventories of key foodstuffs and participatory land use mapping.	Food systems characterized by short chains: Local food system (Kenya). Production and consumption of food within local household: Domestic-indigenous food system (Bolivia). Creation of network of producers, processors, and consumers for food production in and around a city: Agroecological food system (Bolivia).	Human health: Local and agroecological food systems have perceived positive health impacts by producers, workers, and consumers. Agroecological food system has a better capacity to provide what is considered to constitute a “good diet”. Food security: Local, domestic indigenous and agroecological food systems score higher on food security than agro-industrial food system. Environment: Local and agroecological food systems have a higher capacity to provide agroecosystem services. Domestic indigenous food system has the best environmental performance.

## Data Availability

Not applicable.

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
