# Peer review of "A Rapid Review of Territorialized Food Systems and Their Impacts on Human Health, Food Security, and the Environment"

_nutrients, 2021, doi:10.3390/nu13103345_

Round 1
Reviewer 1 Report
I would like to thank the authors for the efforts made to produce such an interesting manuscript. The review is an important contribution to the literature to reach food security in light of all the challenges our society is facing.
The paper is well organised and well written, the methodology is consolidated and well described. I only would like to understand some aspects of this methodology that are fundamental in a systematic review:
- what is the primary research question fundamental for a review?
- what is the timespan of the databases search?
- commenting on table 1, I know that the breakdown of the research question generates different keywords, which are the basis to agree on a unique search term to be used on different databases. In table 1, instead, includes 3 different search terms. Could you please explain better why?
l
Reviewer 2 Report
Dear Authors,
your work is an interesting first step towards getting to know the relevant bibliography.
However, it appears to be of little significance, having taken into consideration only 6 papers.
In any case, it is not well explained how we got from 2207 articles to 36, to then get to select the 6 articles considered.
I would explain it better in the paragraph on sampling.
It would also be appropriate that paragraphs 3.1.1 and following were directly the Results.
In the Discussions I would like your comments to be further argued with other bibliography.
In the Conclusions it would be appropriate to resume and answer the questions posed in the introduction and then continue with your conclusion.Â
Round 2
Reviewer 2 Report
Dear authors thank you for accepting my recommendationsÂ